# Information Theory Opens New Dimensions in Experimental Studies of Animal Behaviour and Communication

**DOI:** 10.3390/ani13071174

**Published:** 2023-03-26

**Authors:** Zhanna Reznikova

**Affiliations:** 1Institute of Systematics and Ecology of Animals, Siberian Branch RAS, Frunze 11, 630091 Novosibirsk, Russia; zhanna@reznikova.net; 2V. Zelman Institute for Medicine and Psychology, Novosibirsk State University, Pirogova 2, 630090 Novosibirsk, Russia

**Keywords:** computational ethology, information theory, Shannon entropy, Kolmogorov complexity, behavioural patterns, classification, animal communication, ants, rodents

## Abstract

**Simple Summary:**

An information theory approach provides new methods for experimental studying and analysis of animals’ communication and behavioural sequences. One of the main problems in studying animal behaviour at different levels of the organisation, from individual to collective behaviour, is searching for a reliable criterion for evaluating the variability and complexity of patterns. The data-compression method based on ideas of Kolmogorov complexity allows us to spot regularities that are difficult to detect otherwise, and that can influence the complexity of behavioural sequences. The information-theory approach allows for a comparative analysis of ethograms and provides a quantitative way to distinguish between innate and experienced behaviour. This approach is focused on studying natural communications by measuring information transmission rates without attempting to decode signals. It has made it possible, in particular, to discover the existence of a developed symbolic “language” in leader-scouting ant species, based on the ability of these ants to transfer abstract information about remote events.

**Abstract:**

Over the last 40–50 years, ethology has become increasingly quantitative and computational. However, when analysing animal behavioural sequences, researchers often need help finding an adequate model to assess certain characteristics of these sequences while using a relatively small number of parameters. In this review, I demonstrate that the information theory approaches based on Shannon entropy and Kolmogorov complexity can furnish effective tools to analyse and compare animal natural behaviours. In addition to a comparative analysis of stereotypic behavioural sequences, information theory can provide ideas for particular experiments on sophisticated animal communications. In particular, it has made it possible to discover the existence of a developed symbolic “language” in leader-scouting ant species based on the ability of these ants to transfer abstract information about remote events.

## 1. Introduction

Animal ethology, going back to the fundamental works of Konrad Lorenz (1903–1989) [1], Niko Tinbergen (1907–1988) [2], and Karl von Frisch (1886–1982) [3], is a multifaceted discipline, which includes many aims and scopes, from the characterisation of behaviour, typically of intact, freely moving animals in their natural environment, to experiential studying animals’ decision-making, problem-solving, social interactions, and communication. Over the last 40–50 years, ethology has become increasingly quantitative and computational. The new field of computational ethology is made possible by advances in technology, mathematics, and engineering that allow scientists to automate the measurement and analysis of animal behaviour [4] and communication [5]. Many impressive results based on “big behavioural data” [6,7,8] have been obtained in studying many domains of ethology, from deciphering “body language” in mice [9] to classifying drawings created by orangutans [10], and from discovering rules regulating social food exchanges within an ant colony [11] to unveiling the possibilities for studying social interactions within primate societies based on face recognition using deep learning [12]. Thus, there are many fields of biology where big data are gathered and analysed successfully, and co-adapting approaches from different fields can open new horizons here. However, eschewing the purely big data approach, where behavioural data are acquired blindly from large numbers of animals through automation and without regard for the individual [4], the organismal level study may lead to insights into both developmental and evolutionary processes and, subsequently, to computational principles shared across species [13]. Comparative studies of communication in human- and non-human-primates can serve as an example here. Until recently, the animal capacity to generate communicative sequences was considered highly constrained in terms of structural systematical and meaning generation [14]. Girard-Buttoz et al. [15] recently revealed a highly versatile vocal system in chimpanzees based on their capacity to organise single units into structured sequences. Although the authors used 4826 recordings of 46 wild chimpanzees, larger datasets are still needed to prove animals’ capacity to produce flexible vocal sequences that support numerous differentiated meanings. At the same time, a developmental study on ten individual marmoset monkeys revealed vocal turn-taking behaviour with similar patterns of phase-locking and entrainment as in human communication [16]. Of course, these studies are not comparable as they set different goals and objectives; this example only illustrates the differences in methodological approaches.

When analysing animal behavioural sequences, researchers’ main problem is finding an adequate model that would allow for assessing certain characteristics of these sequences while using a relatively small number of parameters. In this review, I demonstrate that the information theory approach can furnish effective tools to analyse and compare animal behaviours and communications, especially in cases where researchers have a restricted amount of data. The main idea behind this approach is to consider behavioural patterns as fragments of natural “texts” where the “text” is a sequence of letters coding behavioural acts similar to the sequences of nucleotides in DNA chains or literary and musical texts [17]. In addition to a comparative analysis of behaviour, the information theory approach can provide ideas for particular experiments opening windows to sophisticated animal “languages” [18,19]. The first part of this review concentrates on applying information theory to the comparative analysis of stereotyped behaviours that enable animals to accomplish particular goals, such as chasing prey, avoiding a predator, or defending territories. The method of analysing ethological texts based on ideas of Kolmogorov complexity can serve many goals, from distinguishing between stereotyped and flexible parts of behaviours to understanding the evolutionary roots of complex stereotypes within animal lineages, such as rodents’ families [17,20,21,22]. The second part considers studying animal intelligent communication using an experimental approach based on Shannon entropy and Kolmogorov complexity [18,19,23].

## 2. Applying Ideas of Information Theory for Comparative Analysis and Classification of Ethological Texts

Animal behaviour consists of movements and events, and ethologists must organise their observational data into discernible categories. Charles Otis Whitman (1842–1910) [24] and, a few years later, Oskar Heinroth (1871–1945) [25] created the concept of species-specific elementary behaviour units. For example, in Heinroth’s terminology, the whole sequence of a falcon’s hunting raid (searching, recognising the prey, separating an individual from the flock, diving, and catching) comprises one single “species-specific drive action.” Lorenz [26] restricted its use to characterise the single, overt motor act. Returning to the example of falcon hunting, Lorenz [27] used this term only for the final link in the chain, the actual catching of the prey that he later called a “fixed action pattern” [1]. Since Altmann’s [28] review, behavioural sampling techniques have become increasingly popular [29,30]. Behavioural reactions of animals seem changeable and somewhat ephemeral; however, since the classic works of Heinroth [25] and Lorenz [26,27], behavioural patterns serve as a criterion for distinguishing between species, often as reliable as morphological features. One of the main problems in studying animal behaviour at different levels of organisation, from individual to collective behaviour, is finding a reliable criterion for evaluating the variability and complexity of patterns. The concept of complexity of animal behaviour is still mainly intuitive. First, one must distinguish between the complexity of flexible and stereotypic behaviour. In this aspect, the former refers to the levels of complexity of problems to solve and decisions to make [31], whereas the latter is about the internal coordination and regularity of species-specific repertoire. The more “complex” a behavioural pattern is, the more difficult it is to detect regularities and predict sequences of elements. The information theory approach allows for finding regularities in ethological texts [17].

### 2.1. Terms and Notions

This section aims to give a system of terms and notions that I use today. The terms used in this paper are mostly consistent with those used in referenced works. However, different, and sometimes contradictory, terms are used in the ethological literature, so a comparison and a clarification are in order.

Fixed action pattern (FAP), according to Lorenz [27], is a fixed sequence of stereotyped acts. Since Hailman [32] demonstrated the influence of subtle forms of experience through his investigations of pecking in newly hatched seagulls, we know that motor learning is involved in the emergence of many behaviours typical of a species. The notion of FAP was challenged by posing the question of “how fixed is a fixed action pattern” [33].

The terms and notions based on the revisited concept of FAP [34] are as follows: behavioural elements are minimal units of behaviours, that is, elementary movements and postures, and the ethogram is a comprehensive list, inventory, or description of the behaviour of an organism [27]. *Fixed action pattern* (FAP) is a complex motor act involving a specific temporal sequence of unlearned repetitive movements, which is triggered by an external stimulus and is influenced by underlying motivation, but relatively unaffected by feedback. *Modal action patterns* (MAP) are similar to FAP, but include individual variation in performance of standard behavioural sequence, thus considering FAP as a somehow “open program” [35,36].

Candidate behavioural modules have been recognised in a variety of different contexts and on a wide range of spatio-temporal scales, and, accordingly, researchers have given them a diverse set of names, including motor primitives, behavioural motifs, motor synergies, prototypes, and movements (review in [9]). Gomez-Marin et al. [6] suggest ethomes, sub-ethomes, and ethons to describe levels of behaviours. By analogy to birdsong, Wiltschko et al. [9] propose to refer to the modules identified as behavioural “syllables” and consider the statistical interconnections between these syllables as behavioural “grammar.” Recently, neurobiologists and ethologists have applied the term *stereotyped motor patterns* [37] to describe targeted sequences of stereotyped actions. Alternatively, ethologists use the term *behavioural pattern* in a general sense to denote identifiable types of behaviour, including patterns with a socially learnt origin [38].

Reznikova et al. [34] apply the term *behavioural stereotype*, referring to relatively stable and recurrent sequences of behavioural elements. The term “stereotype” can be confused with the unrelated term *stereotypies*: abnormal behaviour described, for example, in animals confined to zoos or farms, when animals lack control over important aspects of their environment [39]. Reznikova et al. [34] suggest the term *behavioural tuplet* to denote a relatively stable and recurrent chain of behavioural elements. Both behavioural sequences and behavioural stereotypes may include one or more FAPs as parts. These authors define a *behavioural sequence* as an arbitrary sequence of behavioural acts, such as the whole sequence of a falcon’s hunting raid mentioned before as an example. An individual’s *behavioural phenotype* is a combination of its unique behavioural propensities and its responsiveness to environmental variation, also known as behavioural plasticity [40].

### 2.2. Using Ideas of Shannon Entropy and Kolmogorov Complexity for Comparative Analysis of Animals’ Behaviours and Communications

At the end of the 1940s, C. Shannon developed the basis of information theory [41]. The fundamental measure in information theory is entropy, which corresponds to the uncertainty associated with a signal. The foundational role of this theory was appreciated immediately, not only in the development of the technology of information transmission, but also in the study of natural communication systems. It is natural to use information theory in the investigation of communication systems because this theory presents general principles and methods for developing effective and reliable communication systems [18]. In particular, in the 1950s and 1960s, the entropy (degree of uncertainty and diversity) of most European languages was estimated [42,43] (review in [44]). Perhaps, Alan Turing [45] was the first connecting intelligence and computation through an imitation game (review in [46]). Shannon [47] urged caution about extending the engineer’s concept to biology and psychology, but he was also optimistic that the theory would prove useful. As G.A. Miller [48] gave this, “… a few psychologists, whose business throws them together with communication engineers, have been making considerable fuss over something called “information theory,” They drop words like “noise,” “redundancy,” or “channel capacity” into surprising contexts and act like they had a new slant on some of the oldest problems in experimental psychology.” In particular, Frick and Miller [49] used the sequential interpretation to analyse the behaviour of rats in a Skinner box during preconditioning, conditioning, and extinction. They applied the redundancy parameter T to measure stereotypy in animals’ behaviour. Miller [50] addressed several phenomena that seem to exhibit information capacity limitations, including absolute judgments of unidimensional and multidimensional stimuli and short-term memory. Attneave [51] and Barlow [52] (see also [53]) incorporated into their models of visual perception the statistical methods of signal processing and information theory. Wilson [54] applied ideas of Shannon entropy to estimate the quantitative parameters of the honeybee’s ability to memorize the location of a food source. Despite the long legacy of information theory in experimental behavioural research, as the mathematical psychologist Duncan Luce has written [55], information theory has never performed a central role in psychology. However, the data-analytical use of information theory has applications in studies of behavioural psychology (see review in [56]) Information derived concepts have been exploited by behavioural researchers to account for animals’ behaviour from neurons to cognitive mechanisms (review in [57]). Recently, Zenil et al. [58] explored algorithmic connections between animal behaviour, molecular biology, and Turing computation based on ideas of Kolmogorov complexity.

Currently, information theory has many applications for studying the organisational complexity of behaviours, such as signal activities and communication within groups [59,60,61,62,63,64,65], division of labour and information transfer within ant communities [17,18,66], modifications of behaviour under stress conditions [53], switching the degree of randomness in the sequences of rats’ choices during interactions with virtual competitors [67], alterations of escape trajectories and predator evasion abilities in rodents [68], and so on. Schreiber [69] introduced *transfer entropy* to detect asymmetries in the interaction of two coupled dynamical systems from their time series. Since then, we have seen applications of transfer entropy spawning in the most disparate fields of science and engineering, where the identification of cause-and-effect relationships is required (review in [70]). Based on animal-robot experiments studying zebrafish social behaviour and fear response, Porfiri [70] has demonstrated the potential of transfer entropy to assist in detecting and quantifying causal relationships in animal interactions. Following the same approach, Valentini et al. [71] provided evidence that the communication protocol used by leaders and followers over intermediate time scales explains the functional differences between the tandem runs of ants and termites despite their using similar signalling mechanisms at short time scales. It turned out that the bidirectional flow of information is present only in ants and not in termites, and is consistent with the use of acknowledgement signals to regulate the flow of directional information. It is worth noting that, since only one nestmate is recruited at a time, tandem running has been considered a primitive form of recruitment, which, however, is prevalent across the ant phylogeny and thus popular among ant studies (review in [72]).

In comparative ethological studies, the prevalent method is the analysis of ethograms, that is, recordings of behavioural sequences of letters from an alphabet that consists, on average, of 10–15 symbols or letters, each corresponding to a certain behavioural element (an act) [2,73]. When studying communications, such as acoustic signals, researchers identify the distinct units of an acoustic sequence [63]. One can analyse ethograms and communication sequences as biological texts, that is, as sequences of symbols from a finite alphabet.

Sequences of symbols from a finite alphabet (or texts) appear as primary objects of study in many scientific fields, including molecular biology, genetics, linguistics, zoosemiotics, and others. As mentioned in the introduction, the most popular approaches based on the description of sequences by stochastic processes meet some methodological limitations. The main problem is finding an adequate model that would allow for assessing specific characteristics of a biological text while using a relatively small number of parameters. Kolmogorov complexity (after Kolmogorov [73]) furnishes many useful tools for studying different natural processes that can be expressed using sequences of symbols from a finite alphabet. V. Zelman Institute for

Medicine and Psychology, [73,74]. Formally, it is the length of a shortest program from which the text can be reconstructed [75]. It is worth noting that quantitative estimation of the complexity of sequences in natural texts is of interest in its own right [17]. A huge body of literature analyses symbolic sequences by means of Kolmogorov complexity, including diagnostic of the authorship of literary and musical texts (reviews in [17,22]). The measure is based on Kolmogorov complexity when evaluating the complexities of texts in different languages. It measures the information content of a string by the length of the shortest possible description required to (re)construct the exact string [76]. Although Kolmogorov complexity is incomputable, it can be conveniently approximated with text compression programs. The basic idea behind the compression technique is that text samples, which can be compressed comparatively better are linguistically comparatively less complex [77]. The importance of the use of ideas and methods of Kolmogorov complexity in biological sciences can hardly be overestimated. Such methods were used for estimation of closeness of DNA sequences [78,79], phylogenetic tree building [80,81], networks of interactions and relationships in rodents [82], evaluation of information entropy of humpback whale songs [83], and so on. The data compression method suggested by Ryabko and colleagues [17] is based on the concept of Kolmogorov complexity and allows to search for regularities within sequences of symbols with relatively small numbers of parameters. The main idea behind the data compression method is that it is able to capture all kind of regularities in the text and does so in a way amenable to formal statistical analysis.

To analyse ethograms, one should encode each behavioural element by a separate letter. The “regularity” means any characteristic of a text that makes it more predictable, such as repeated subsequences, letters, or sequences that are more likely to occur when preceded or followed by certain other letters or subsequences, and so on. In general, regularities may be complex and involve arbitrary computable functions, but when applying a data compressor, we restrict ourselves to those regularities it can capture. When comparing ethograms of different species as biological texts, the method implies the presence of regularities within behavioural patterns and finds all differences in the complexity of the ethograms. If one sees such differences, additional biological data are needed to explain them. An ideal data compressor would be able to capture all possible regularities in a text and thus compress it to its Kolmogorov complexity. An actual data compressor can capture only some regularities—those that the algorithm behind the data compressors is designed to capture. Nevertheless, this typically goes beyond the frequencies of individual letters, or all words of a given length, which is what is analysed by Markov models. Whatever the data compressor, one can reason precisely about the outcome of the resulting test. Specifically, the probability of type I error (that the difference in complexities is found whereas there is none) is guaranteed to be less than the pre-specified confidence level. It is worth mentioning that some other specific regularities of biological texts have been studied previously. For example, T-pattern analysis [84,85] makes it possible to reveal recurring sequences of events that are not necessarily consecutive, allowing for a deeper analysis of the structure of the behaviour, including its temporal characteristics. However, no formal and rigorous statistical tests are available based on these regularities. As distinct from other approaches to analysing sequences using Kolmogorov complexity, the considered method stays within the framework of mathematical statistics.

When comparing ethograms (see [20]), the researchers consider the two following hypotheses: *H*_0_ = {the sequences from both sets are generated by one source} and *H*_1_ = {the sequences from the different sets are generated by stationary and ergodic sources with different Kolmogorov complexities per letter of generated sequences}. Specifically, this can be completed as follows: (1) from the sequences to be compared fragments (*x*_1_…*x*_t_) of equal length *t* are selected randomly so that the Mann–Whitney–Wilcoxon test can be applied to the resulting fragments; (2) the complexity of each fragment is defined as *K*(*x*_1_…*x*_t_) = |*φ*(*x*_1_…*x*_t_)|/*t*, where *φ* is a data compressor, and |*φ*(*x*_1_…*x*_t_)| is the length of the fragment of the sequence compressed by the data compressor; and (3) applying the Mann–Whitney–Wilcoxon test, we test the hypothesis *H*_0_. To test the hypotheses, we should represent the sequence of symbols as text files. Then, these text files should be compressed by the chosen data compression method. The level of compression corresponds to the ratio between the length of the file after and before the compression. The difference between compression ratios of files to be compared reflects the difference between complexities of the symbol sequences recorded. Thus, one can use the compression ratio as a characteristic of complexity (see details in [17,20]). The ability of different data compressors to compress information is highly dependent on the chosen method of compression, that is, on the algorithm used to find regularities in the file to be compressed. There are many lossless data compressors applicable to texts. In [20], the authors review various data compressors and justify their research choice of compressor BZip2. It is worth noting that using a weak data compressor, that is, a one that can spot fewer regularities, results in a lower power of the test. It means that in such a case, *H*_0_ can be chosen where *H*_1_ should have been; however, the opposite probably cannot happen with a probability higher than the pre-specified level, no matter how bad the data compressor is.

To test the method’s validity, Ryabko and co-authors [17] considered ethological texts of two kinds: hunting behaviours of ants and territorial behaviours of gulls. In the first case, the researchers explored an ant behavioural model. The hunting stereotype of ants *Myrmica rubra* towards jumping springtails includes determining the victim, approaching it, and performing the “tip-and-run attack” (FAP). The ant attacks the prey, bends the abdomen and head to the thorax, jumps towards the springtail, falls on it abruptly, and stings [86]. Rearing ants from pupae in the laboratory enabled researchers to reveal that ants possess innate behavioural programs of different completeness. Those specimens equipped with the complete pattern enjoy an inherited template of potential prey and a readily available hunting stereotype; others possess fragments of the pattern and complete them through simple forms of social learning [17,34]. Ryabko et al. [17] compared two groups of highly genetically variable ants: members of a natural colony (“wild”) and naive (laboratory-reared from pupae) ants. They represented behavioural sequences as texts in which behavioural units (10 in total), singled out from video records and denoted by symbols (letters), served as an alphabet and tested the Hypothesis *H*_0_ (the sequences from two sets are generated by one source) against *H*_1_ (the complexity of sequences from one set is, on average, larger than the complexity of sequences from the other). As the starting point of a hunting stereotype, the authors took the ant’s approach to the victim and the display of purposive movements; transportation of the killed victim was considered the end of the complete stereotype. All cases of loss of the victim and switching to another one were considered ends of incomplete stereotypes. It turned out that complete (successful) hunting stereotypes in members of a natural ant colony are characterised by smaller complexity than incomplete hunting stereotypes in naive laboratory-reared ants. Together with similar results obtained later on rodents [21,34], this application of Kolmogorov complexity could help to distinguish between innate and learned components of behavioural stereotypes.

In the second case, the researchers applied the “resident–intruder” experimental paradigm to compare territorial behaviours in gulls *Larus ridibundus* in two situations: (1) a mocking intruder is approaching a nest in which a gull sits on the eggs, and (2) an intruder is moving away from the nest. The reactions of a resident towards an approaching intruder appeared to be more variable and “chaotic” than its reactions towards an escaping one, because, in the first situation, a resident hectically tries various ways to drive the trespasser away. In contrast, in the second situation, it simply repeats successful combinations of FAPs. There are some similarities with recent results obtained in the damselfish [87] and the swimming crab [88].

The suggested method for studying animal behavioural patterns is a promising tool to be used in different behavioural and evolutionary research areas. In particular, this method can help extract “basic” (completely innate) behavioural patterns by comparing different levels of complexity and flexibility of behavioural sequences without rearing naive animals. By analysing the complexity of behavioural patterns in naive and experienced animals, researchers can link the experience with structure and function. This is particularly important for evolutionary studies, including behavioural mechanisms of speciation.

Reznikova and co-authors [20,21] applied the data compression method to compare the organisational complexity of species-specific hunting stereotypes in nine small mammal species with various diets and different levels of phylogenetic relationship. In their laboratory experiments, eight Muroidea species with different diets, mainly herbivorous, displayed skilful attacks towards the live insect prey typical for actively pursuing predators, such as the grasshopper mouse of the genus *Onychomys* (see details in [34]). Unlike *Onychomys*, the studied hamsters, voles, and mice species lack any morphological and physiological adaptations. The fact that a variety of herbivorous and granivorous species have preserved an optional hunting pattern in their repertoire makes this behaviour a good candidate for studying the evolution of hunting behaviour within rodents’ lineage (Figure 1). The data compression method has proven to be an effective tool for the comparative analysis of behaviours between and within species and groups of individuals. This method allows us to spot regularities that are difficult to detect otherwise and can influence the complexity of behavioural sequences. Then, the authors proceed with searching for explanations of similarities and differences in behaviours. To find differences between hunting stereotypes in various rodent species and the insectivorous “standard,” the authors compared levels of complexities in hamsters, voles, and mice species with those of the common shrew. Seven rodent species studied displayed succinct, highly predictable hunting stereotypes, making it easy for the data compressor to find regularities. With its changeable manipulation of prey and less predictable transitions between stereotype elements, the generalist Norway rat significantly differed from other species. The complexities of hunting stereotypes in young and adult rats are similar, and both groups had no prior experience with the prey. One can conjecture that it is not learning but rather the specificity of the organisation of the stereotype that is responsible for the nature of the hunting behaviour in rats. Reznikova et al. [21] speculate that rodents possess different hunting behaviours, one of which is based on a succinct insectivorous standard, and another type, perhaps characteristic of generalists, which is less ordered and is characterised by poorly predictable transitions between elements.

### 2.3. Using Data-Compressors for Classification Behavioural Sequences: A New Tool to Evaluate Species Differences

As described in the previous section, the data-compression method is sensitive to the presence of regularities within texts representing behavioural sequences and effective to measure their relative complexity. However, this method does not help to determine similarities in behavioural strategies of animals. One of the main problems in comparative studies of animal behaviour is searching for a decisive tool for evaluating the similarities and differences between behaviour patterns, especially within closely related taxa. The primary rationale for the use of phylogenetically based statistical methods is that phylogenetic signal, the tendency for related species to resemble each other, is ubiquitous; however, behavioural traits exhibit a lower signal than other features, such as morphological, life-history, or physiological ones [89,90]. Classification of similar behavioural sequences in different species would help to understand the relationship between behavioural plasticity and evolutionary processes within a definite lineage. The solution to these problems depends to a great extent on the availability of an adequate mathematical method. As described in the previous section, the data compression method already revealed a surprising similarity between the hunting behaviours of the insectivorous common shrew and several rodent species, mainly herbivorous and granivorous. However, further behavioural observations showed that the hunting modes could differ in different species. The differences concern the order of particular behavioural elements and some aspects of hunting attacks in various species. So, although rodent species display similar predictability of transitions between elements within behavioural sequences, they possibly possess dissimilar structures of hunting behaviour [91]. Thus, we need a new tool to evaluate differences between the structural features of the ethological “texts.”

Ryabko and co-authors [92] suggested a compression-based solution for classification of ethological texts. This method belongs to the framework of mathematical statistics and allows pairwise comparisons. In the case of ethogram comparison, this approach allows quantifying the degree of structural similarities and differences between behavioural sequences as ethological “texts.” Levenets et al. [22] developed this method to evaluate structural differences between hunting behaviours in nine species of small mammals with various ecological traits and different diets. The authors recorded hunting behaviour towards an insect in individual members of eight rodent species and the insectivorous shrew as a “standard predator.” The authors tested the hypothesis of whether the behavioural sequences of different species as “texts” are generated either by a single source or by different ones. The main idea of the approach is to combine fragments of the behavioural sequence of one species (“text X”) with fragments of another one (“text Y”) and then compress the combined sequences by an archiver. The text files which contain similar sequences will be compressed better. The authors applied the open-source data compressor 7-zip v. 18.05 (64-bit), which uses the method of data compression called Bzip2, (compressed file format bz2). Preliminary, the researchers compared three data compression algorithms, namely, LZMA, Deflate, and BZip2, and chose the one that compressed better (see details in [22]). Based on the association coefficients obtained from pairwise comparisons, the researchers built a new classification of types of hunting behaviours, which brought a unique insight into how hunting behaviour in rodents possibly changed and evolved. In particular, the behavioural sequences of the shrew *Sorex araneus* and the tuvinian vole *Alticola tuvinicus* appeared to differ from those of all other species. From the ethological point that the herbivorous vole, such as the insectivorous shrew, differs from the rest of the species enables us to search for distinct traits in its hunting attacks. Interestingly, four hamster species ended up in the same cluster as the rat *Rattus norvegicus*. The fact that all hamster species bear similarities confirm the validity of the method. That precisely the sequences of *R. norvegicus*, *Allocricetulus eversmanni,* and *Al. curtatus*, are generated by one source, although hamsters and rats are not phylogenetically close, possibly caused by the particular abilities of these three species to manipulate with forepaws when handling the prey. It is of specific interest that recently, Levenets et al. [91] revealed similarities between these two hamster species and the Norway rat at a behavioural level. The new method enabled the researchers to find quantitative confirmation of these similarities.

## 3. The Use of Ideas of Information Theory for Studying “Language” and Intelligence in Animals: An Insight from Leader-Scouting Ant Species

The fundamental question about the uniqueness of human language still needs to be understood. Evolutionary linguistics has emerged as an attempt to answer questions concerning language evolution and human nature based on an interdisciplinary collaboration [93,94,95]. A possible path forward suggested by Hauser et al. [96] includes observations and experiments of naturally communicating animals and experiments assessing animals’ computational and perceptual capacities, focusing on the abilities necessary for human language processing.

Three main experimental paradigms exist in studying animal language behaviour. The first approach is aimed at direct decoding the function and meaning of animal signals, which is a notoriously tricky problem. Two types of natural messages decoded up to the present concern the symbolic honeybee “dance language” deciphered by Karl Von Frisch [3,97], and fragments of acoustic communications in several species, such as monkeys ([98]; review in [99]) and dolphins [100]. As noted in the introduction, a newly discovered highly versatile vocal system in chimpanzees [15] needs larger datasets to prove apes’ abilities to produce flexible vocal sequences that support numerous differentiated meanings. Studying animals’ grammatical and syntax abilities can help us to understand the gap between human linguistic rule-learning skills and those of non-humans (see for example [62,101,102,103]). Recently, Sainburg and co-authors [64] analysed the sequential dynamics of songs from multiple songbird species and speech from multiple languages by modelling the information content of signals as a function of the sequential distance between vocal elements and revealed functionally equivalent dynamics governed by similar processes. The second approach, which is based on artificial intermediary languages (“language-training experiments”), uncovered significant “linguistic” and cognitive potential in some species, that contrasts limitations in understanding their natural communications [104,105,106]; see also reviews in [31,107,108]. The third approach applies ideas of information theory to studying animal communication.

The first attempt to quantify information in animal communication was made by Haldane and Spurway in 1954 [109] on the base of von Frisch’s [110] discovery of honeybee’s dance language. The authors applied the theory of the measurement of information, as developed particularly by Shannon [41] and Weaver [111], and suggested a new field of “ethological cybernetics.” The authors noted that it is not possible to make an accurate measurement of the amount of information concerning distance given by a bees’ dance, because we do not know what, if anything, a group of bees would do in response to a dance giving no such information. Now, we have a better understanding of what and how well the bees indicate with their dance (see review in [98,112]). Schürch and Ratnieks [113] using their own calibration data showed that the direction component conveys 2.9 bits and the distance component 4.5 bits of information, which agrees to some extent with Haldane and Spurway’s estimates that were based on data gathered by von Frisch.

There are two aspects of the information theory approach to studying animal language behaviour: the use of information theory to assess the diversity, complexity, and development of communicative repertoires (see, for example [60,63,114,115]), and the experimental way to studying communications based on Shannon entropy [41] and Kolmogorov complexity [73], which I analyse in this review.

The main point of the experimental approach proposed by Reznikova and Ryabko [18,116] is to study natural communications and evaluate their capabilities by measuring information transmission rates (details in: [19,23]). Since it avoids the need to study the nature of the signals or to decipher messages, this approach provides a different way for understanding the essentials of animal communication systems.

Leader-scouting ant species turn out to be even better candidates for studying general communication rules than the iconic honeybee. There is a challenging problem of understanding the nature of leadership and how consensus emerges from the properties of social networks in animal societies, from primates to ants [117]. In ants, there are two well-studied cooperative tasks which require information transferring: house-hunting and group foraging [118]. A detailed analysis of the ant literature [72,119] shows that mass recruiting and tandem running systems employed by most ant species do not display substantial lifelong distinction among different foraging roles in individuals. Since task fidelity is weak in all these species, there is still no evidence of individual differences between leaders and followers [120]. Even in group-recruiting species, with their targeted communication between recruiters and recruited nestmates, there is only slight, if any, evidence of ants’ careers and behavioural consistency as leaders (for details, see [121]). Personal traits characterise groups of individuals at the colony level, but not performers of functional roles during group-recruiting foraging. Among about 15,000 ant species, the majority display relatively simple modes of communication, and only a few members of the *Formica rufa* group (“red wood ants;” subgenus *Formica s. str*.) possess the sophisticated leader-scouting system based on a consistent personal difference between scouting and foraging individuals within personalised teams. Unique life-history features of *Formica s. str.* favour cognitive specialisation within their societies. This specialisation may be based on the ability of certain individuals to learn faster within specific domains. Red wood ants’ abilities to encode sequences of turns along their way to a goal [72,107] can be considered a cognitive adaptation to their particular foraging style within the tree crown. The leader-scouting seems to be the only foraging system based on consistent personal differences between scouting and foraging individuals [72,121,122], resembling in some traits the “star” network model described by Sueur et al. [117].

To reveal the power of ants’ “language,” the researchers applied two central notions of information theory, that is, (1) the quantity of information, and (2) the duration of time spent by the agents for transmitting one bit. The crucial idea is that the experimenters know exactly the quantity of information to be transferred. This approach, which is based on the “binary tree” experimental paradigm [19], enabled the authors to estimate the rate of information transmission in ants and to reveal that these intelligent insects can grasp regularities in the “texts” (such as LRRL, where L is “left” and R is “right”) and use them for coding and “compression” of information. These abilities may be considered the most complex properties of ants and animal cognition, and communication in general [31]. The other series of experiments on “counting ants” was based on the Shannon’s equation connecting the length of a message (*l*) and its frequency (*p*), i.e., *l* = −*log p*, for rational communication systems. Applying this concept, Reznikova and Ryabko [23] demonstrated that ants could transfer information about the number of objects to each other and even add and subtract small numbers to optimise their messages. Below, I explain some details of these experiments.

A general scheme of all three series of long-term experiments (“binary tree,” “counting,” and “arithmetic”) was based on the necessity to transfer distant information from scouting ants to members of their foraging groups to obtain food (see details in [7,8,37,90]). Each laboratory ant colony lived in an arena, in a transparent nest that made it possible for their activity to be observed. The arena was divided into two sections: the smaller one containing the nest, and the bigger one with an experimental system (Figure 2). To prevent access to the food in the maze by a straight path, the set-up was placed in a bath of water. All actively foraging ants were individually marked with paint. In each series or experiments, several hundred of scouting ants and their foraging groups took part. For example, 335 scouts and their teams from three *F. polyctena* colonies were used in the experiments with the binary tree. The scouts and their teams took part in different trials switching from one task to another. In each trial, the experimenter placed one of the scouts on a definite point of a maze, for example, on a leaf of the binary tree (or a branch of a counting maze) that contained a trough with the food, then it returned to the nest by itself. There were no cues that could help the ants find the food (including olfactory ones) except the information contacts with scouts (see the detailed description, tables and figures in [72,107,122]). When a scout returned home, the experimenter measured the duration of its contact with foragers in seconds.

As noted before, the idea of the first series of experiments is that the experimenters know the quantity of information (in bits; [41]) to be transferred, which corresponds to the number of turns in a “binary tree” maze towards a “leaf” containing syrup. The duration of the contacts between the scouts and foragers appeared to be *ai* + *b*, where *i* is the number of turns, *a* is the time duration required for transmitting one bit of information, and *b* is an introduced constant. The rate of information transmission was about 1 min per bit in ants, which is at least 10 times smaller than in humans languages [123,124]. Ants appeared to be able to grasp regularities in the “text” (sequence of turns, see below) to be transferred and use them to “compress” and, thus, optimize their messages. The idea of experiments on “information compression” in ants was inspired by the concept of Kolmogorov complexity [73] applied to words (or “texts”) composed of the letters of an alphabet, for example, consisting of two letters: L (left) and R (right) corresponding to a sequence of turns in a “binary tree” maze. As noted before, informally, the Kolmogorov complexity of a word (and its uncertainty) equates to its most concise description. For example, the word “LRLRLRLR” can be represented as “4LR,” while the “random” word of shorter length “LRRLRL” probably cannot be expressed more concisely, and this is more complex. The hypothesis being tested was *H*_0_, that is, the time for transmission of information by the scout does not depend on the complexity of the “text.” The alternative hypothesis was *H*_1_ that this time actually depends on the complexity of the “text.” The hypothesis *H*_0_ was rejected (*p* = 0.01), thus showing that the more time ants spent on the information transmission, the more complex—in the sense of Kolmogorov complexity—was the message [18]. This surprisingly resembles “learning with chunking and generalization” during the foraging process in structured environments suggested by Kolodny et al. [125].

The series of experiments on “counting ants” suggests an original experimental paradigm based on information theory. Perhaps, no field of cognitive ethology is based on comparing human and non-human abilities to such a great extent as the field of studying reasoning about numbers. One of the key findings over the past decades is that our number faculty is deeply rooted in our biological ancestry, and not based on our ability to use language [126]. It is worth noting that the degree to which non-verbal organisms display “counting” (“absolute numerosity discriminations”) is still controversial. The term “proto-counting,” suggested by Davis and Pérusse [127], refers to instances of counting-like behaviour of animals where not all of the “counting principles” are met (reviews In [23,126,128]). Until recently, all experimental paradigms for investigating numerical processing in animals are restricted by studying subjects at the individual level (reviews in [23,126]). Reznikova and Ryabko [23,116] suggested an experimental scheme for studying numerical competence in ants based on their communications, which is still the first to exploit the natural communicative systems of animals [31,107,129]. In this respect we encounter a methodological paradox. It is evident that high levels of number-related skills are closely connected with language development in humans. At the same time, all known experimental paradigms for studying numerical competence in animals do not exploit the phenomenon of close relations between intelligence, sociality, and natural communication. Even in the honey bee studies [112,130,131,132,133], cognitive capacities of these social insects have been tested individually, and the capabilities of honey bees’ extraordinary symbolic language [3] were not used in the experimental schemes.

In the series of experiments on “counting ants,” scouting individuals were required to transfer to foragers in the laboratory nest the information about which branch of a particular “counting maze” they had to go to obtain syrup. “Counting maze” is a collective name for several variants of set-ups (see the detailed description, tables, and figures in [19,23]). The main idea of this experimental paradigm is that experimenters can judge how ants *represent* numbers by estimating how much time individual ants spend on “pronouncing” numbers, that is, on transferring information about the index numbers of branches. The researchers used the comb–like set-ups: a “horizontal trunk,” a “vertical trunk,” and a “circle”, and compared the duration of information contacts between scouts and foragers, which preceded successful trips by the foraging teams.

The relation between the index number of the branch (*j*) and the duration of the contact between the scout and the foragers (*t*) appeared to be well described by the equation *t* = *c j* + *d* for different set-ups, which are characterized by different shapes, distances between the branches, and lengths of the branches. The values of parameters *c* and *d* are close and do not depend either on the lengths of the branches or on other parameters. Interestingly, the quantitative characteristics of the ants’ “number system” seem to be close, at least outwardly, to some archaic human languages: the length of the code of a given number is proportional to its value. For example, the word “finger” corresponds to 1, “finger, finger” to the number 2, “finger, finger, finger” to the number 3 and so on [134,135]. Of course, this is nothing more than a superficial analogy. The numerical meaning attached to fingers is culturally encoded and in strikingly diverse ways [136]. In modern human languages, the length of the code word of a number *j* is approximately proportional to *log j* (for large *j*), and the modern numeration system results from a long and complicated development. However, the concept of a linear number line appears to be a cultural invention that fails to develop in the absence of formal education [135] (see review in [137]).

The degree to which animals are capable of mental arithmetic reflects their capacity to mentally transform numerical information [138]. Some findings suggest that arithmetic capacity is rooted deeply in evolutionary history. Rugani and colleagues [139] found arithmetic abilities in newly hatched domestic chicks. Honeybees appeared to learn to use blue and yellow as symbolic representations for addition or subtraction [133]. Again, the distinction of the experimental paradigm for studying arithmetic skills in ants is the use of animals’ natural communications. The main idea is to exploit the fundamental statement of information theory known as the Zipf’s law [140]: in a “reasonable” communication system, the frequency of usage of a message and its length must correlate. The informal pattern is quite simple: the more frequently a message is used in a language, the shorter the word or the phrase coding it [141]. This phenomenon is manifested in all known human languages [142] and recently has been analysed in animal communication [115].

The main experimental procedure in this “arithmetic” series of experiments was similar to other experiments with “counting mazes” [19,23] Four colonies of red wood ants took part in experiments over three years, with the number of trials ranging from 92 to 150 in different years. Ants were offered a horizontal trunk with 30 branches. The experiments were divided into three stages, with various regularity of placing the reward on branches with different numbers. In the first stage, the branch containing the reward was selected randomly, with equal probabilities for all units. So the probability of finding the reward on a particular branch was 1/30. Thus, the first stage of the “arithmetic” experiments did not differ from “counting” experiments described above. In the second stage, two “special” branches, A and B (N 7 and N 14; N 10 and N 20; and N 10 and N 19 in different years), contained the syrup much more frequently than the rest of the branches: with probability of 1/3 for “A” and “B,” and 1/84 for each of the other 28 units. In this way, two “messages” that scouting ants should transfer to members of their foraging teams (“the trough is on branch A” and “the trough is on branch B”) had a much higher probability than the remaining 28 messages. In one series of trials, there was only one “special” point A (branch N 15). On this branch, the food appeared with probability of 1/2 and 1/58 for each of the other 29 units. In the third stage of the experiment, the branch with the trough was chosen uniformly random again.

The obtained data demonstrated that ants appeared to be forced to develop a new code in order to optimize their messages, and the usage of this new code has to be based on simple arithmetic operations. The patterns of dependence of the information transmission time on the number of the food-containing branch at the first and third stages of experiments were considerably different. In the vicinities of the “special” branches, the time taken for transmission of the information about the number of the branch with the trough was, on the average, shorter. For example, in the first series, at the first stage of the experiments the ants took 70–82 s to transmit the information about the fact that the trough with syrup was on branch N 11, and 8–12 s to transmit the information about branch N 1. At the third stage it took 5–15 s to transmit the information about branch N 11. Analysis of the time duration of information transmission by the ants raises the possibility that at the third stage of the experiment the scouts’ messages consisted of two parts: the information about which of the “special” branches was the nearest to the branch with the trough, and the information about how many branches away is the branch with the trough from a certain “special” branch. In other words, the ants, presumably, passed the “name” of the “special” branch nearest to the branch with the trough, then the number which had to be added or subtracted in order to find the branch with the trough. That ant teams went directly to the “correct” branch enables the researchers to suggest that they performed correctly whatever “mental” operation (subtraction or addition) was to be made (see details in [23,107]). It is likely that at the third stage of the experiment the ants used simple additions and subtractions, achieving economy in a manner reminiscent of the Roman numeral system when the numbers 10 and 20, 10 and 19 in different series of the experiments performed a role similar to that of the Roman numbers V and X. This also indicates that these insects have a communication system with a great degree of flexibility. Until the frequencies with which the food was placed on different branches started exhibiting regularities, the ants were “encoding” each number (j) of a branch with a message of length proportional to j, which suggests unitary coding. Subsequent changes of code in response to special regularities in the frequencies are in line with a basic information-theoretic principle that in an efficient communication system the frequency of use of a message and the length of that message are related (see also the review in [58]).

To conclude this section, leader-scouting ants can (1) transfer information, (2) compress information, (3) change the way they represent information, and (4) add and subtract small numbers. First, when applied to ants, this experimental paradigm can extend to highly social vertebrates and thus open new horizons in studying numerical cognition. It can be suggested that the scheme of experiments based on solving cooperative tasks can be directly used on some non-human mammals and young human infants. For example, pairs or small groups of subjects can be involved. In this case, one subject is shown the container with reward, but has to communicate its location to the other for both of them to get it. The reward can be placed in 1 out of (say) 20 containers, which perform the same role as the branches of the comb-like maze in the experiments with ants. The numerical abilities of the subjects can then be studied by analysing the time the first subject (that has seen the reward) spends on transferring the information on its location to the other, precisely as in the experiments with ants.

## 4. Conclusions

The information theory approach can serve as an effective tool to analyse animal behavioural patterns and communications, opening a window to sophisticated animal “languages.” The review considers two ways of applying this approach: analysing ethograms as biological texts and creating new experimental paradigms for studying communications and underpinning cognitive capacities in non-human animals.

The data-compression method of analysing behavioural sequences as ethological texts based on ideas of Kolmogorov complexity can serve many goals, from distinguishing between stereotyped and flexible parts of behaviours to understanding the evolutionary roots of complex stereotypes within animal lineages. The compression-based method for testing and classification of behavioural sequences belongs to the framework of mathematical statistics. It allows one to compare the structural characteristics of any text in pairwise comparisons. Based on association coefficients obtained from pairwise comparisons, it became possible to build a new classification of hunting behaviours in rodents. This brought insight into how particular elements of predatory patterns changed and evolved.

The experimental approach based on Shannon entropy and Kolmogorov complexity gives some insights into experimental paradigms for studying natural communications and evaluating their capabilities by measuring information transmission rates without investigating the nature of the signals and attempts to decipher messages. A long-term experimental study on ant communication and intelligence revealed the insects’ ability to transfer abstract information about remote events and estimate the information transmission rate. The obtained results are essential not only for biology, but also for cognitive science, linguistics, cybernetics, and robotics.

## Figures and Tables

**Figure 1 animals-13-01174-f001:**
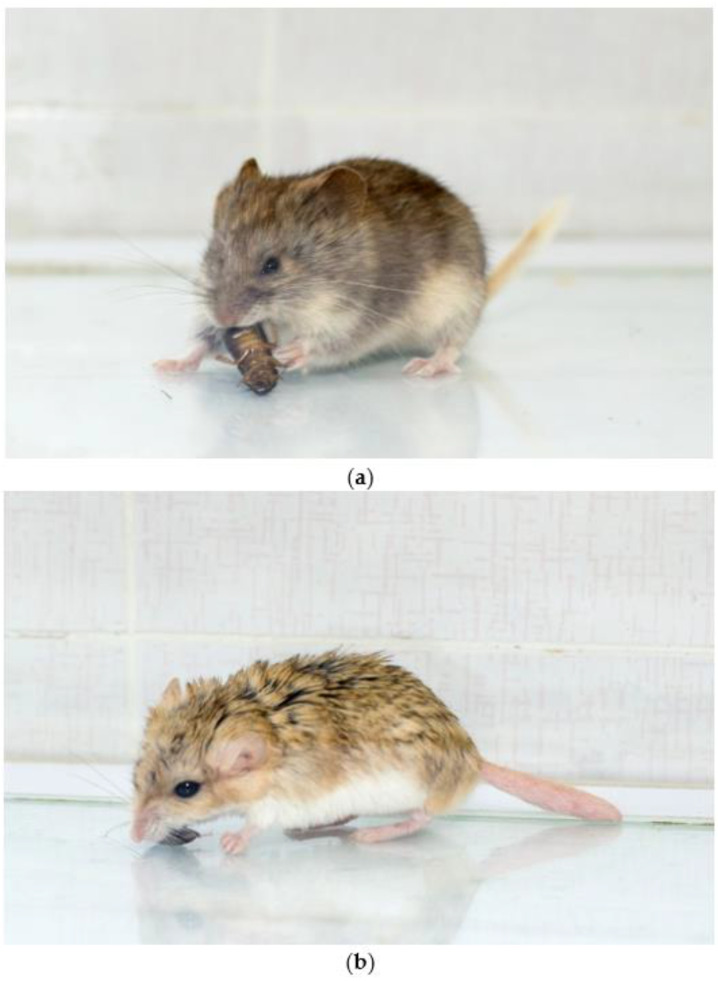
(**a**) a narrow-headed vole *Lasiopodomys gregalis* is handling the cockroach with its paws; (**b**) a fat-tailed gerbil *Pachyuromys duprasi* is attacking the prey. Photos made by Galina Azarkina.

**Figure 2 animals-13-01174-f002:**
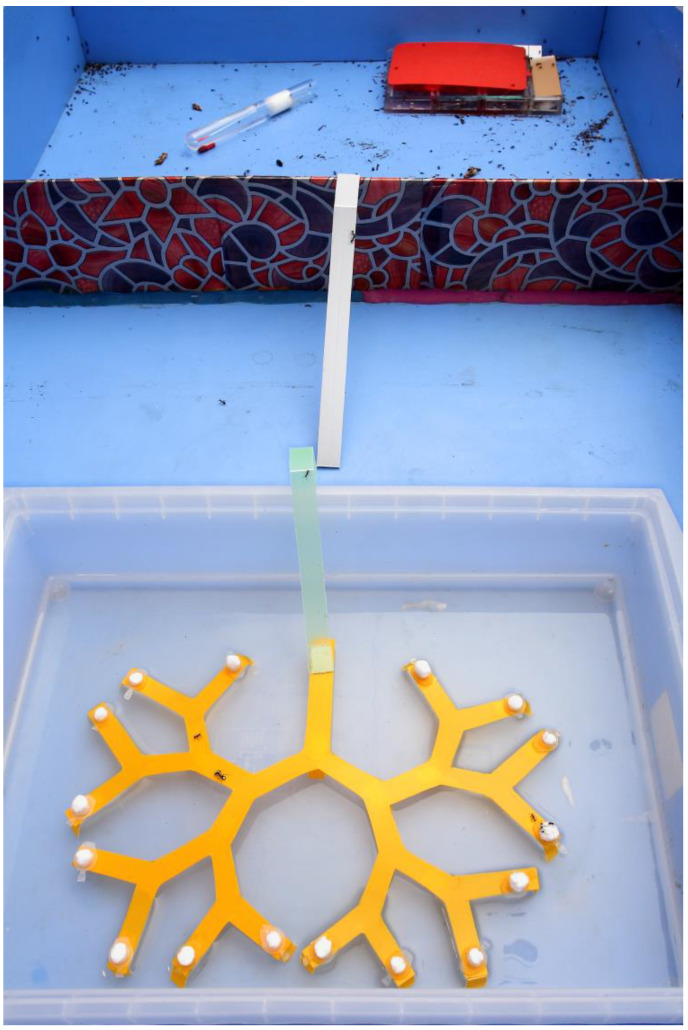
A laboratory arena divided into two parts, containing an artificial ant nest and a binary tree maze placed in a bath with water. This binary tree has four forks. Photo taken by Nail Bikbaev.

## Data Availability

Not applicable.

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
