# Peer review of "Information Theory Opens New Dimensions in Experimental Studies of Animal Behaviour and Communication"

_animals, 2023, doi:10.3390/ani13071174_

Round 1

Reviewer 1 Report (Previous Reviewer 1)

Dear Author,

Thank you for submitting this paper that explores information theory for review. I have reviewed a previous version of this paper that was submitted through Animals, and so have seen the initial review comments that were raised.

My main concern is that the majority of the comments raised on initial review have still not been addressed. For example, there remain excessive self-citations in the work. Some of the citations have in fact just been shuffled - it appears there are more self-citations in this work than previously. This shows quite a biased review of the field given the sheer number of self-citations. Please remove some of these citations and read widely if your work is to be reconsidered.

Please also include scientific names for all species when they are first mentioned in the text - for example mouse (Mus musculus). This was commented on at initial review, and so I am concerned that this has not been addressed. 

Above all, the explanations of Shannon and Kolmogorov are still seriously confused, leading to an unclear background to your study. Please provide a much clearer explanation of these concepts if resubmitting this paper.

Author Response

Dear Reviewer,

I am grateful for the thorough analysis and valuable comments for the previous version that helped me to improve the manuscript. Here are my answers to your concerns about the current version.

Yours sincerely,

Zhanna Reznikova

Concern :

My main concern is that the majority of the comments raised on initial review have still not been addressed. For example, there remain excessive self-citations in the work. Some of the citations have in fact just been shuffled - it appears there are more self-citations in this work than previously. This shows quite a biased review of the field given the sheer number of self-citations. Please remove some of these citations and read widely if your work is to be reconsidered.

Answer.

Following your suggestion to remove some self-citations in the previous version, I removed two of them from a reference list. However, although the number of self-references decreased, there are even more self-citations in the MS.

Some of the citations to my own work have been added in response to the comments by the reviewers who requested more explanation. While I can understand that the paper might seem heavy on self-citation, its reference list reflects the current state of the art of this sub-field of science and is based on a careful review of the current literature, as well as on my expertise built over several decades of working in this area. Furthermore, every reference is in place for a reason and cannot be removed without affecting the text's integrity.

Concern :

Please also include scientific names for all species when they are first mentioned in the text - for example mouse (Mus musculus). This was commented on at initial review, and so I am concerned that this has not been addressed.

Answer:

Concerning specifically the ‘mouse’ example, I found only one mention of mouse in the text : “the grasshopper mouse of the genus Onychomys ”. That won’t be true to name Onychomys Mus musculus because these are two different species.

All other species mentioned in the text have their scientific names, for example, “... behavioural sequences of the shrew Sorex araneus and the tuvinian vole Alticola tuvinicus”, That precisely the sequences of R. norvegicus, Allocricetulus eversmanni and Al.curtatus,…” and so on.

Such mentions as “ From the ethological point that the herbivorous vole, like the insectivorous shrew, differs…” should not include their scientific names because this is not their first mention. 

Such mentions as, for example, “….from deciphering “body language” in mice [9] to classifying drawings created by orangutans [10] and from discovering rules regulating social food exchanges within an ant colony” refer to collective references, in this case, to several species of mice, orangutans and ants. In these cases, the scientific names are mentioned in the referenced papers. Likewise, in the phrase “Unlike Onychomys, the studied hamsters, voles and mice species lack any morphological and physiological adaptations.” collective species names are used. This is a standard scientific practice to use collective names of species.

Concern:

Above all, the explanations of Shannon and Kolmogorov are still seriously confused, leading to an unclear background to your study. Please provide a much clearer explanation of these concepts if resubmitting this paper.

Answer:

I think the background information provided in the current version is adequate for understanding the paper, given also its overall length and scope. Moreover, these are classical concepts, so that the reader interested in a more in-depth explanation can find any number of textbooks and tutorials by means of a simple Web search.

Reviewer 2 Report (Previous Reviewer 3)

This is a resubmitted manuscript by Reznikova on the benefits of information theory on studying animal behaviour and communication. I had a few conceptual comments and suggestions on the manuscript, which the author has either followed or argue convincingly when she rather not. I think the manuscript has improved a lot in quality and can be published now

A couple of typos to correct:

Line 295: provably -> probably

Line 501: reference 112 repeated

Author Response

Dear Reviewer,

Thank you for your careful consideration of my paper. I am grateful for the thorough analysis and valuable comments that helped me to improve the manuscript during the first round. In the current version, I corrected the misprints you mentioned.

Yours sincerely,

Zhanna Reznikova

Reviewer 3 Report (Previous Reviewer 4)

In this revised manuscript, the author has done a really good job addressing the concerns raised in the previous review round. I would like to thank the author for taking the points raised into account and for carefully revising their manuscript. It is much easier to read and follow now. The author now provides adequate background information and accounts of the long legacy of Information Theory in behavioral research, which was missing from the previous version. All the other issues that I had previously pointed out have been clarified. I have no additional comments to the revised manuscript and feel my previous comments have been addressed adequately.

Author Response

Dear Reviewer,

Thank you for your careful consideration of my paper. I am grateful for the thorough analysis and valuable comments that helped me to improve the manuscript.

Yours sincerely,

Zhanna Reznikova

This manuscript is a resubmission of an earlier submission. The following is a list of the peer review reports and author responses from that submission.

Round 1

Reviewer 1 Report

Dear Authors,

Thank you for submitting this paper that explores Information theory in animal behavioural research. This is an interesting paper with some potential value to researchers. There are some interesting points made in the review – though the real world use of the findings could be developed further.

There are some revisions required in order to consider this manuscript for publication. I have included specific feedback on the PDF document version of the manuscript, please find attached. Additionally, please address the following key areas when making revisions:

1.      Real-world application. At current, the use of this theory for practitioners is difficult to pull out of the paper. How should they code behaviour? What does this mean in terms of statistical analysis? Please consider these points more carefully so that this is a functional document for researchers to make use of.

2.      Consider citations – I can see 18 references to the author’s existing work which is somewhat surprising. Some of the points made in order to include all these references are dubious. Please ensure that the wider work is read and clearly cited instead of these self-citations.

3.      Make sure scientific names are consistently included in the work.

With these revisions, the paper should be in a better position overall.

Author Response

Dear Reviewer,

Thank you for your careful consideration of my paper. I am grateful for a thorough analysis and valuable comments that helped me to improve the manuscript. All the edits suggested in the review have been implemented.

Yours sincerely,

Zhanna Reznikova

Rev 1 Comments and Suggestions for Authors

Dear Authors,

Thank you for submitting this paper that explores Information theory in animal behavioural research. This is an interesting paper with some potential value to researchers. There are some interesting points made in the review – though the real world use of the findings could be developed further.

There are some revisions required in order to consider this manuscript for publication. I have included specific feedback on the PDF document version of the manuscript, please find attached. Additionally, please address the following key areas when making revisions:

1.      Real-world application. At current, the use of this theory for practitioners is difficult to pull out of the paper. How should they code behaviour? What does this mean in terms of statistical analysis? Please consider these points more carefully so that this is a functional document for researchers to make use of.

Answer. I added the following fragments:

lines 72-75: The main idea behind this approach is to consider behavioural patterns as fragments of natural “texts” where the “text” is a sequence of letters coding behavioural acts similar to the sequences of nucleotides in DNA chains or literary and musical texts [17]

lines 273 – 297: When comparing ethograms (see [20]), the researchers consider the two following hypotheses: H0 = {the sequences from both sets are generated by one source} and H1 = {the sequences from the different sets are generated by stationary and ergodic sources with different Kolmogorov complexities per letter of generated sequences}. Specifically, this can be done as follows: (1) from the sequences to be compared fragments (x1...xt) of equal length t are selected randomly so that the Mann–Whitney–Wilcoxon test can be applied to the resulting fragments; (2) the complexity of each fragment is defined as K(x1...xt) = |φ(x1...xt)| / t, where φ is a data compressor, and |φ(x1...xt)| is the length of the fragment of the sequence compressed by the data compressor; (3) applying the Mann–Whitney–Wilcoxon test, we test the hypothesis H0. To test the hypotheses, we should represent the sequence of symbols as text files. Then these text files should be compressed by the chosen data compression method. The level of compression corresponds to the ratio between the length of the file after and before the compression. The difference between compression ratios of files to be compared reflects the difference between complexities of the symbol sequences recorded. So one can use the compression ratio as a characteristic of complexity (see details in [17,20]). The ability of different data compressors to compress information is highly dependent on the chosen method of compression, that is, on the algorithm used to find regularities in the file to be compressed. There are many lossless data compressors applicable to texts. In [20], the authors review various data compressors and justify their research choice of compressor BZip2. It is worth noting that using a weak data compressor, that is, a one that can spot fewer regularities, results in a lower power of the test. It means that in such a case, H0 can be chosen where H1 should have been; however, the opposite provably cannot happen with a probability higher than the pre-specified level, no matter how bad the data compressor is.

2.      Consider citations – I can see 18 references to the author’s existing work which is somewhat surprising. Some of the points made in order to include all these references are dubious. Please ensure that the wider work is read and clearly cited instead of these self-citations.

Answer. To meet this concern, I revisited the MS's reviewing parts, referring to much broader work and shortened self-citations a bit. The literature cited adequately represents the current development in this particular area.

(1) lines 155 – 210. At the end of the 1940s, C. Shannon developed the basis of information theory [41]. The fundamental measure in information theory is entropy, which corresponds to the uncertainty associated with a signal. The foundational role of this theory was appreciated immediately, not only in the development of the technology of information transmission, but also in the study of natural communication systems. It is natural to use information theory in the investigation of communication systems because this theory presents general principles and methods for developing effective and reliable communication systems [18]. In particular, in the 1950s and 1960s the entropy (degree of uncertainty and diversity) of most European languages was estimated [42,43] (review in [44]). Perhaps, Alan Turing [45] was the first connecting intelligence and computation through an imitation game (review in [46]). Shannon [47] urged caution about extending the engineer’s concept to biology and psychology, but he was also optimistic that the theory would prove useful. As G.A. Miller [48] gave this, “...a few psychologists, whose business throws them together with communication engineers, have been making considerable fuss over something called “information theory,” They drop words like “noise,” “redundancy,” or "channel capacity" into surprising contexts and act like they had a new slant on some of the oldest problems in experimental psychology.” In particular, Frick and Miller [49] used the sequential interpretation to analyse the behaviour of rats in a Skinner box during preconditioning, conditioning, and extinction. They applied the redundancy parameter T to measure stereotypy in animals’ behaviour. including absolute judgments of unidimensional and multidimensional stimuli and short-term memory. Attneave [51] and Barlow [52] (see also [53]) incorporated into their models of visual perception the statistical methods of signal processing and information theory. Wilson [54] applied ideas of Shannon entropy to estimate the quantitative parameters of the honeybee’s ability to memorize the location of a food source. Despite the long legacy of information theory in experimental behavioural research, as the mathematical psychologist Duncan Luce has written [55], information theory has never played a central role in psychology. However, the data-analytical use of information theory has applications in studies of behavioural psychology (see review in [56]) Information-derived concepts have been exploited by behavioural researchers to account for animals’ behaviour – from neurons to cognitive mechanisms (review in [57]).Recently, Zenil et al. [58] explored algorithmic connections between animal behaviour, molecular biology and Turing computation based on ideas of Kolmogorov complexity.

Currently, information theory has many applications for studying the organisational complexity of behaviours, such as signal activities and communication within groups [59-65], division of labour and information transfer within ant communities [17,18,66], modifications of behaviour under stress conditions [53], switching the degree of randomness in the sequences of rats’ choices during interactions with virtual competitors [67], alterations of escape trajectories and predator evasion abilities in rodents [68], and so on. Schreiber [69] introduced transfer entropy to detect asymmetries in the interaction of two coupled dynamical systems from their time series. Since then, we have seen applications of transfer entropy spawning in the most disparate fields of science and engineering, where the identification of cause-and-effect relationships is required (review in [70]). Based on animal-robot experiments studying zebrafish social behaviour and fear response, Porfiri [70] has demonstrated the potential of transfer entropy to assist in detecting and quantifying causal relationships in animal interactions. Following the same approach, Valentini et al. [71] provided evidence that the communication protocol used by leaders and followers over intermediate time scales explains the functional differences between the tandem runs of ants and termites despite their using similar signalling mechanisms at short time scales. It turned out that the bidirectional flow of information is present only in ants, and not in termites and is consistent with the use of acknowledgement signals to regulate the flow of directional information. It is worth noting that, since only one nestmate is recruited at a time, tandem running has been considered a primitive form of recruitment, which, however, is prevalent across the ant phylogeny and thus popular among ant studies (review in [72]).

(2) lines 455 – 467:

The first attempt to quantify information in animal communication was made by Haldane and Spurway in 1954 [109] on the base of von Frisch’s [110] discovery of honeybee’s dance language. The authors applied the theory of the measurement of information, as developed particularly by Shannon [41] and Weaver [111] and suggested a new field of “ethological cybernetics.” The authors noted that it is not possible to make an accurate measurement of the amount of information concerning distance given by a bees’ dance, because we do not know what, if anything, a group of bees would do in response to a dance giving no such information. Now we have a better understanding of what–and how well–the bees indicate with their dance (see review in [98,112]). Schürch and Ratnieks [113] using their own calibration data showed that the direction component conveys 2.9 bits and the distance component 4.5 bits of information, which agrees to some extent with Haldane and Spurway’s estimates that were based on data gathered by von Frisch.

3.      Make sure scientific names are consistently included in the work.

Answer: I did this

Reviewer 2 Report

Small formal correction ideas for the author:

There are several names of historic persons, like, for example, Fibonacci. Please give also their first name and in parenthesis the date of birth and the date of death, so that the reader has immideately an idea who was it.

Line 108: Replace “Tviebhandlung” by the correct German term “Triebhandlung”.

Line 113: “.” after “)”.

Line 190: The “(“ is never closed.

Line 372: The “(“ is never closed.

Line 382: The “(“ is never closed.

Line 402: “A detailed analysis of ant literature shows that mass recruiting and tandem running systems employed by most ant species do not display substantial lifelong distinction among different foraging roles.”

Are individuals meant herer? Should be claryfied.

Line 411: What should “a few Formica rufa species” mean? There is only one Formica rufa species. Is the Formica rufa complex meant, or the Formica rufa group? See Seifert (2021) for definitions of taxa within the wood ants.

Line 422: “Ants have always been helping people to solve various problems. Everybody remembers how they sorted seeds for Cinderella.”

Is this a joke? I don’t know why ants have always been helping people to solve various problems…

Line 428: The “(“ is never closed.

It is often written about “animal and human abilities” etc. Isn’t it enough to write about “animal abilities”? Human are animals.

What is about the study of Cammaerts & Cammaerts (2021, Behavioral Sciences 11: Ants can anticipate the following quantity in an arithmetic sequence)? Shouldn’t it be cited? Does the author not believe in their results?

Many inconsistences in the References (italics, points, etc.), some examples:

Line 738: Put scientific name in italics.

Line 742: Replace “ten Cate” by “Ten Cate”.

Line 747: Put scientific name in italics.

Line 768: Put scientific name in italics.

Line 780: Put scientific name in italics.

Line 782: Replace “?.” by “?”.

Line 782: Endpoint missing.

Author Response

Dear Reviewer,

Thank you for your careful consideration of my paper. I am grateful for a thorough analysis and valuable comments that helped me to improve the manuscript. All the edits suggested in the review have been implemented.

Yours sincerely,

Zhanna Reznikova

Comments and Suggestions for Authors

Small formal correction ideas for the author:

Concern:

There are several names of historic persons, like, for example, Fibonacci. Please give also their first name and in parenthesis the date of birth and the date of death, so that the reader has immideately an idea who was it.

Answer: I removed the mention of Fibonacci, and added information about some historical persons

Line 108: Replace “Tviebhandlung” by the correct German term “Triebhandlung”.

Answer: I removed this term

Line 113: “.” after “)”. + +

Line 190: The “(“ is never closed.+

Line 372: The “(“ is never closed.+

Line 382: The “(“ is never closed.+

Line 402: “A detailed analysis of ant literature shows that mass recruiting and tandem running systems employed by most ant species do not display substantial lifelong distinction among different foraging roles.”

Are individuals meant herer? Should be claryfied.

Answer: I added most ant species do not display substantial lifelong distinction among different foraging roles in individuals.

Line 411: What should “a few Formica rufa species” mean? There is only one Formica rufa species. Is the Formica rufa complex meant, or the Formica rufa group? See Seifert (2021) for definitions of taxa within the wood ants.

Answer: I changed this sentence as follows : only a few members of the to the Formica rufa group (“red wood ants”; subgenus Formica s.str. )

Line 422: “Ants have always been helping people to solve various problems. Everybody remembers how they sorted seeds for Cinderella.”

Is this a joke? I don’t know why ants have always been helping people to solve various problems…Answer: I removed this sentence.

Line 428: The “(“ is never closed.+

It is often written about “animal and human abilities” etc. Isn’t it enough to write about “animal abilities”? Human are animals.

What is about the study of Cammaerts & Cammaerts (2021, Behavioral Sciences 11: Ants can anticipate the following quantity in an arithmetic sequence)? Shouldn’t it be cited? Does the author not believe in their results?

Answer:

I replaced “ animal and human abilities” with “human and non-human abilities”. As far as the question about citing the study of Cammaerts & Cammaerts (2021), the “formal” answer is that I consider the experimental scheme of studying ants’ numerical competence based on information transmission, and in this aspect, the mentioned Myrmica study is not relevant. My “informal” answer is that I had not entered into details of Cammaerts & Cammaerts’s studies, especially since their report on self-recognition in Myrmica, which I did not find to be credible.

Many inconsistences in the References (italics, points, etc.), some examples:

Answer: I corrected it in the text

Line 738: Put scientific name in italics.

Line 742: Replace “ten Cate” by “Ten Cate”. +

Line 747: Put scientific name in italics.+

Line 768: Put scientific name in italics.

Line 780: Put scientific name in italics.

Line 782: Replace “?.” by “?”.

Line 782: Endpoint missing.

Reviewer 3 Report

Manuscript ID: Animals- 2082033

In this manuscript, Reznikova reviews the use of information theory on studying communication and behaviour in animals, largely based on her own and her colleagues’ the work. Large datasets on animal behavior and communication have recently been made available by automated observations and technological advances and Reznikova proposes information theory as an alternative for analysing them. I have mixed feelings about this manuscript, but I have to admit that I was not very familiar with the information theory itself before reading this manuscript. On one hand, the manuscript shows well the many uses and the benefits of information theory. However, while the manuscript is generally well-written and though out, there are some issues that need to be justified better. It also seems a bit incoherent, dealing partly with issues that seem irrelevant and partly leaving some crucial details out. I’m not sure how familiar the readers of this special issue are with complex theoretical issues, such as information theory, and with demanding quantitative methods, but I suggest that the manuscript is revised and condensed on these lines.

Major points

When reading this manuscript, I’m getting a feeling that the author’s only reason for promoting information theory and data compression is the fear of vast amount of data and problems analysing it. I would put my money on modern quatitative methods and the superior computing power we enjoy today, however. There are many fields of biology, where big data are gathered and analysed successfully. My intuition also says that rather than making existing but hard-to-find patterns clearer, data may be lost when compressed. This should be clarified and justified.

The author gives a list of biological fields (line 192-4) where Kolmogorov complexity has been used, but states that it does not allow hypothesis testing and thus limits its use. I don’t quite understand the logic here. Surely similarity of DNA sequencies, phylogenetic trees and interaction networks can be used as hypothesis and those are not always tested with conventional statistics, but, for instance, by Bayesian interference.

In many places, the ideas how information theory is applied to studying animal behavior is given quite succinctly, and I believe that a regular reader would struggle understanding it. For instance, without any prior knowledge of how data compression is actually done, I was surprised to see how poorly it was explained (line 325-330). I still have no idea how it allows to find existing but hard-to-find patterns, how it is done, how the three algorithms mentioned work and why BZip2 is the best.

Minor comments

It would be good to mention early on that this paper is a review.

I don’t see a reason for defining terms and notions in length like now (line 125- 163). Terms such as FAP and MAP are not much used elsewhere in the manuscript or not at all and could be defined when used.

When there is a single author, the use of plural (us) seems odd.

The writing is mostly quite good, but unnecessarily convoluted in places. What additional values does it bring to mention Nobel prize winners (twice) or Fishers statistical hypothesis testing? It just makes the text longer. There are also some odd sentences and minor slips. For instance, “anything which can be put in a computer” (line 185) should be expressed in a more formal way. Line 268: word “predator” is repeated.

Line 231. maybe accidentally instead of luckily?

Author Response

Dear Reviewer,

Thank you for your careful consideration of my paper. I am grateful for a thorough analysis and valuable comments that helped me to improve the manuscript. All the edits suggested in the review have been implemented.

Yours sincerely,

Zhanna Reznikova

Comments and Suggestions for Authors

Manuscript ID: Animals- 2082033

In this manuscript, Reznikova reviews the use of information theory on studying communication and behaviour in animals, largely based on her own and her colleagues’ the work. Large datasets on animal behavior and communication have recently been made available by automated observations and technological advances and Reznikova proposes information theory as an alternative for analysing them. I have mixed feelings about this manuscript, but I have to admit that I was not very familiar with the information theory itself before reading this manuscript. On one hand, the manuscript shows well the many uses and the benefits of information theory. However, while the manuscript is generally well-written and though out, there are some issues that need to be justified better. It also seems a bit incoherent, dealing partly with issues that seem irrelevant and partly leaving some crucial details out. I’m not sure how familiar the readers of this special issue are with complex theoretical issues, such as information theory, and with demanding quantitative methods, but I suggest that the manuscript is revised and condensed on these lines.

Major points

Concern:

When reading this manuscript, I’m getting a feeling that the author’s only reason for promoting information theory and data compression is the fear of vast amount of data and problems analysing it. I would put my money on modern quatitative methods and the superior computing power we enjoy today, however. There are many fields of biology, where big data are gathered and analysed successfully. My intuition also says that rather than making existing but hard-to-find patterns clearer, data may be lost when compressed. This should be clarified and justified.

Answer.

My informal answer is that the methods of information theory are not designed for big data analysis but instead for the cases where the sequences are relatively short and it is not possible to estimate a large number of parameters. Only lossless data compressors were used in the studies reviewed here. To clarify, I added the following fragments:

(1) lines 48-50:

So, there are many fields of biology where big data are gathered and analysed successfully, and co-adapting approaches from different fields can open new horizons here.

(2) lines 67 - 77 : When analysing animal behavioural sequences, researchers' main problem is finding an adequate model that would allow for assessing certain characteristics of these sequences while using a relatively small number of parameters. In this review, I demonstrate that the information theory approach can furnish effective tools to analyse and compare animal behaviours and communications, especially in cases where researchers have a restricted amount of data. The main idea behind this approach is to consider behavioural patterns as fragments of natural “texts” where the “text” is a sequence of letters coding behavioural acts similar to the sequences of nucleotides in DNA chains or literary and musical texts [17].Besides a comparative analysis of behaviour, the information theory approach can provide ideas for particular experiments opening windows to sophisticated animal “languages” [18,19].

Concern:

The author gives a list of biological fields (line 192-4) where Kolmogorov complexity has been used, but states that it does not allow hypothesis testing and thus limits its use. I don’t quite understand the logic here. Surely similarity of DNA sequencies, phylogenetic trees and interaction networks can be used as hypothesis and those are not always tested with conventional statistics, but, for instance, by Bayesian interference.

Answer:

Bayesian inference requires putting prior distributions over biological data, which means making strong modelling assumptions about it. In such applications as behavioural sequences, there are no adequate modelling assumptions. I am not aware of any Bayesian-inference works that would reason about Kolmogorov complexity, which must be for the same reason.

To clarify, I added the following fragment:

(1) lines 228-238: It is worth noting that quantitative estimation of the complexity of sequences in natural texts is of interest in its own right [17]. A huge body of literature analyses symbolic sequences by means of Kolmogorov complexity, including diagnostic of the authorship of literary and musical texts (reviews in [17,22]). The measure is based on Kolmogorov complexity when evaluating the complexities of texts in different languages. It measures the information content of a string by the length of the shortest possible description required to (re)construct the exact string [76]. Although Kolmogorov complexity is uncomputable, it can be conveniently approximated with text compression programs. The basic idea behind the compression technique is that text samples which can be compressed comparatively better are linguistically comparatively less complex [77].

Concern:

In many places, the ideas how information theory is applied to studying animal behavior is given quite succinctly, and I believe that a regular reader would struggle understanding it. For instance, without any prior knowledge of how data compression is actually done, I was surprised to see how poorly it was explained (line 325-330). I still have no idea how it allows to find existing but hard-to-find patterns, how it is done, how the three algorithms mentioned work and why BZip2 is the best.

Answer. To meet this concern, I added the following fragment:

lines 273-297: When comparing ethograms (see [20]), the researchers consider the two following hypotheses: H0 = {the sequences from both sets are generated by one source} and H1 = {the sequences from the different sets are generated by stationary and ergodic sources with different Kolmogorov complexities per letter of generated sequences}. Specifically, this can be done as follows: (1) from the sequences to be compared fragments (x1...xt) of equal length t are selected randomly so that the Mann–Whitney–Wilcoxon test can be applied to the resulting fragments; (2) the complexity of each fragment is defined as K(x1...xt) = |φ(x1...xt)| / t, where φ is a data compressor, and |φ(x1...xt)| is the length of the fragment of the sequence compressed by the data compressor; (3) applying the Mann–Whitney–Wilcoxon test, we test the hypothesis H0. To test the hypotheses, we should represent the sequence of symbols as text files. Then these text files should be compressed by the chosen data compression method. The level of compression corresponds to the ratio between the length of the file after and before the compression. The difference between compression ratios of files to be compared reflects the difference between complexities of the symbol sequences recorded. So one can use the compression ratio as a characteristic of complexity (see details in [17,20]). The ability of different data compressors to compress information is highly dependent on the chosen method of compression, that is, on the algorithm used to find regularities in the file to be compressed. There are many lossless data compressors applicable to texts. In [20], the authors review various data compressors and justify their research choice of compressor BZip2. It is worth noting that using a weak data compressor, that is, a one that can spot fewer regularities, results in a lower power of the test. It means that in such a case, H0 can be chosen where H1 should have been; however, the opposite provably cannot happen with a probability higher than the prespecified level, no matter how bad the data compressor is.

Minor comments

It would be good to mention early on that this paper is a review.

Answer: I did this

I don’t see a reason for defining terms and notions in length like now (line 125- 163). Terms such as FAP and MAP are not much used elsewhere in the manuscript or not at all and could be defined when used.

Answer. to clarify the aim of this section, I added the following fragment: (lines 111-114) : This section aims to give a system of terms and notions as I see today. The terms used in this paper are mostly consistent with those used in referenced works. However, different and sometimes contradictory terms are used in the ethological literature, so a comparison and a clarification is in order.

When there is a single author, the use of plural (us) seems odd.

Answer: I corrected this

The writing is mostly quite good, but unnecessarily convoluted in places. What additional values does it bring to mention Nobel prize winners (twice) or Fishers statistical hypothesis testing? It just makes the text longer. There are also some odd sentences and minor slips. For instance, “anything which can be put in a computer” (line 185) should be expressed in a more formal way. Line 268: word “predator” is repeated.

Answer: I corrected this

Line 231. maybe accidentally instead of luckily?

Answer: I corrected this

Reviewer 4 Report

The author reviews a set of experiments that exploited the Information Theory to explain animal behaviour and communication. In particular, the author focuses on two notions linked to this theory - Kolmogov’s complexity and Shannon’s entropy – and presents their potential application in 4 main areas of animal behaviour: 1) the analysis of regularities within behavioural sequences; 2) the analysis of similarities between behavioural sequence of different species; 3) the communication of spatial information; 4) numerical cognition.

In my opinion, the content of this manuscript is consistent with the past work of the author on this topic (e.g., Ryabko & Reznikova, 2009), however, it is overly self-referential. In its current state, the manuscript does not recognize the long legacy of Information Theory in behavioral research (e.g., Luce, 2003). The lack of relevant background on this topic makes the manuscript hard to follow, at the current state. Before presenting any experiment, I recommend that the author includes a brief paragraph in the Introduction, where the two pivotal concepts of this manuscript (Kolmogorov's complexity and Shannon's entropy) are immediately defined, together with a brief excursus about how/why these information-derived concepts have been exploited by behavioral researchers to account for animals' behavior - from neurons to cognitive mechanisms (e.g., Sayood, 2018).

Below, I listed some other specific points to address:

Line 209: The author should dedicate some lines to describe the compression method more in depth. How does it actually work? What exactly is compressed? What kind of data? How does the compression work? The description is too vague at the moment.

Line 213: “An ideal data compressor would be able to capture all possible regularities in a text and thus compress it to its Kolmogorov complexity”. What exactly is the Kolmogorov complexity of a text? Is it an index? How is it computed?

Lines 242-245: the author compares complete hunting stereotypes of a natural ant colony vs incomplete hunting stereotypes of laboratory-reared ants. What do “complete” and “incomplete” mean in this context? This distinction should be introduced in the text

Line 256: The author claims that this method can identify innate behaviours without mentioning any evidence to support this. Could you please elaborate on this point? What is the link between the complexity of a behaviour and its innateness?

Line 276: As it is described, this method seems sensitive to the presence of regularities/stereotypes within 'texts' representing behavioural sequences. As the author pointed out, it is useful to measure their relative complexity. How can this method help to determine similarities in behavioral strategies of animals, too? Does it concern any analysis of the content (not just the regularity) of the behavioural sequence?

Line 340: “However, hamsters and rats are not phylogenetically close, possibly caused by the particular abilities of these three species to manipulate with forepaws when handling the prey.” Could you please rephrase this line? The subject of the verb “caused” is not clear.

Line 500: “Interestingly, the quantitative characteristics of the ants’ “number system” seem to be close, at least outwardly, to some archaic human languages: the length of the code of a given number is proportional to its value.” Could you please elaborate on this? It is not clear what you mean and which evidence can support this statement.

Line 513: “The main idea here explores the fundamental statement of information theory: in a “reasonable” communication system, the frequency of usage of a message and its length must correlate. The informal pattern is quite simple: the more frequently a message is used in a language, the shorter the word or the phrase coding it. This phenomenon is manifested in all known human languages.” Could you provide any reference to support this?

Line 518: Could you please describe this experiment more accurately? I think the main confusion arises between the words “regularity” and “probability”, “stage” and “series”. It is not clear if different stages comprised different series and trials and if the same individual performed all the stages.  If the same individual performed different trials: “the probability of finding the reward on a particular branch was 1/30”, does it mean that the probability for an individual to find the rewarded branch is 1/30, or that that specific branch was rewarded 1 out of 30 trials? What is a “series” of trials and which stage contained more than one series? If the same individual performed all the stages, then the difference between the three stages can be just a learning effect. I am not doubting the methods of this experiment (which has already passed the peer review). I am just giving to the author some feedback about the general confusion that a potential reader has while reading this part of the manuscript.

Luce RD. Whatever Happened to Information Theory in Psychology? Review of General Psychology. 2003;7:183

Sayood, K. (2018). Information theory and cognition: a review. Entropy, 20(9), 706.

Author Response

Dear Reviewer,

Thank you for your careful consideration of my paper. I am grateful for a thorough analysis and valuable comments that helped me to improve the manuscript. All the edits suggested in the review have been implemented.

Yours sincerely,

Zhanna Reznikova

The author reviews a set of experiments that exploited the Information Theory to explain animal behaviour and communication. In particular, the author focuses on two notions linked to this theory - Kolmogov’s complexity and Shannon’s entropy – and presents their potential application in 4 main areas of animal behaviour: 1) the analysis of regularities within behavioural sequences; 2) the analysis of similarities between behavioural sequence of different species; 3) the communication of spatial information; 4) numerical cognition.

Concern:

In my opinion, the content of this manuscript is consistent with the past work of the author on this topic (e.g., Ryabko & Reznikova, 2009), however, it is overly self-referential. In its current state, the manuscript does not recognize the long legacy of Information Theory in behavioral research (e.g., Luce, 2003). The lack of relevant background on this topic makes the manuscript hard to follow, at the current state. Before presenting any experiment, I recommend that the author includes a brief paragraph in the Introduction, where the two pivotal concepts of this manuscript (Kolmogorov's complexity and Shannon's entropy) are immediately defined, together with a brief excursus about how/why these information-derived concepts have been exploited by behavioral researchers to account for animals' behavior - from neurons to cognitive mechanisms (e.g., Sayood, 2018).

Answer. To meet this concern, I revisited the MS's reviewing parts, referring to much broader work and shortened self-citations a bit. The literature cited adequately represents the current development in this particular area.

(1) lines 155 – 210:

At the end of the 1940s, C. Shannon developed the basis of information theory [41]. The fundamental measure in information theory is entropy, which corresponds to the uncertainty associated with a signal. The foundational role of this theory was appreciated immediately, not only in the development of the technology of information transmission, but also in the study of natural communication systems. It is natural to use information theory in the investigation of communication systems because this theory presents general principles and methods for developing effective and reliable communication systems [18]. In particular, in the 1950s and 1960s the entropy (degree of uncertainty and diversity) of most European languages was estimated [42,43] (review in [44]). Perhaps, Alan Turing [45] was the first connecting intelligence and computation through an imitation game (review in [46]). Shannon [47] urged caution about extending the engineer’s concept to biology and psychology, but he was also optimistic that the theory would prove useful. As G.A. Miller [48] gave this, “...a few psychologists, whose business throws them together with communication engineers, have been making considerable fuss over something called “information theory,” They drop words like “noise,” “redundancy,” or "channel capacity" into surprising contexts and act like they had a new slant on some of the oldest problems in experimental psychology.” In particular, Frick and Miller [49] used the sequential interpretation to analyse the behaviour of rats in a Skinner box during preconditioning, conditioning, and extinction. They applied the redundancy parameter T to measure stereotypy in animals’ behaviour. Miller [50] addressed several phenomena that seem to exhibit information capacity limitations, including absolute judgments of unidimensional and multidimensional stimuli and short-term memory. Attneave [51] and Barlow [52] (see also [53]) incorporated into their models of visual perception the statistical methods of signal processing and information theory. Wilson [54] applied ideas of Shannon entropy to estimate the quantitative parameters of the honeybee’s ability to memorize the location of a food source. Despite the long legacy of information theory in experimental behavioural research, as the mathematical psychologist Duncan Luce has written [55], information theory has never played a central role in psychology. However, the data-analytical use of information theory has applications in studies of behavioural psychology (see review in [56]) Information-derived concepts have been exploited by behavioural researchers to account for animals’ behaviour – from neurons to cognitive mechanisms (review in [57]).Recently, Zenil et al. [58] explored algorithmic connections between animal behaviour, molecular biology and Turing computation based on ideas of Kolmogorov complexity.

Currently, information theory has many applications for studying the organisational complexity of behaviours, such as signal activities and communication within groups [59-65], division of labour and information transfer within ant communities [17,18,66], modifications of behaviour under stress conditions [53], switching the degree of randomness in the sequences of rats’ choices during interactions with virtual competitors [67], alterations of escape trajectories and predator evasion abilities in rodents [68], and so on. Schreiber [69] introduced transfer entropy to detect asymmetries in the interaction of two coupled dynamical systems from their time series. Since then, we have seen applications of transfer entropy spawning in the most disparate fields of science and engineering, where the identification of cause-and-effect relationships is required (review in [70]). Based on animal-robot experiments studying zebrafish social behaviour and fear response, Porfiri [70] has demonstrated the potential of transfer entropy to assist in detecting and quantifying causal relationships in animal interactions. Following the same approach, Valentini et al. [71] provided evidence that the communication protocol used by leaders and followers over intermediate time scales explains the functional differences between the tandem runs of ants and termites despite their using similar signalling mechanisms at short time scales. It turned out that the bidirectional flow of information is present only in ants, and not in termites and is consistent with the use of acknowledgement signals to regulate the flow of directional information. It is worth noting that, since only one nestmate is recruited at a time, tandem running has been considered a primitive form of recruitment, which, however, is prevalent across the ant phylogeny and thus popular among ant studies (review in [72])

(2) lines 455 – 467:

The first attempt to quantify information in animal communication was made by Haldane and Spurway in 1954 [109] on the base of von Frisch’s [110] discovery of honeybee’s dance language. The authors applied the theory of the measurement of information, as developed particularly by Shannon [41] and Weaver [111] and suggested a new field of “ethological cybernetics.” The authors noted that it is not possible to make an accurate measurement of the amount of information concerning distance given by a bees’ dance, because we do not know what, if anything, a group of bees would do in response to a dance giving no such information. Now we have a better understanding of what–and how well–the bees indicate with their dance (see review in [98,112]). Schürch and Ratnieks [113] using their own calibration data showed that the direction component conveys 2.9 bits and the distance component 4.5 bits of information, which agrees to some extent with Haldane and Spurway’s estimates that were based on data gathered by von Frisch.

Below, I listed some other specific points to address:

Line 209: The author should dedicate some lines to describe the compression method more in depth. How does it actually work? What exactly is compressed? What kind of data? How does the compression work? The description is too vague at the moment.

Answer. I added the following fragments:

(1)

line 248-249: To analyse ethograms, one should encode each behavioural element by a separate

letter...

(2) lines 273 – 297: When comparing ethograms (see [20]), the researchers consider the two following hypotheses: H0 = {the sequences from both sets are generated by one source} and H1 = {the sequences from the different sets are generated by stationary and ergodic sources with different Kolmogorov complexities per letter of generated sequences}. Specifically, this can be done as follows: (1) from the sequences to be compared fragments (x1...xt) of equal length t are selected randomly so that the Mann–Whitney–Wilcoxon test can be applied to the resulting fragments; (2) the complexity of each fragment is defined as K(x1...xt) = |φ(x1...xt)| / t, where φ is a data compressor, and |φ(x1...xt)| is the length of the fragment of the sequence compressed by the data compressor; (3) applying the Mann–Whitney–Wilcoxon test, we test the hypothesis H0. To test the hypotheses, we should represent the sequence of symbols as text files. Then these text files should be compressed by the chosen data compression method. The level of compression corresponds to the ratio between the length of the file after and before the compression. The difference between compression ratios of files to be compared reflects the difference between complexities of the symbol sequences recorded. So one can use the compression ratio as a characteristic of complexity (see details in [17,20]). The ability of different data compressors to compress information is highly dependent on the chosen method of compression, that is, on the algorithm used to find regularities in the file to be compressed. There are many lossless data compressors applicable to texts. In [20], the authors review various data compressors and justify their research choice of compressor BZip2. It is worth noting that using a weak data compressor, that is, a one that can spot fewer regularities, results in a lower power of the test. It means that in such a case, H0 can be chosen where H1 should have been; however, the opposite provably cannot happen with a probability higher than the pre-specified level, no matter how bad the data compressor is.

Line 213: “An ideal data compressor would be able to capture all possible regularities in a text and thus compress it to its Kolmogorov complexity”. What exactly is the Kolmogorov complexity of a text? Is it an index? How is it computed?

Answer. I added the following fragment (lines) 228-238:

It is worth noting that quantitative estimation of the complexity of sequences in natural texts is of interest in its own right [17]. A huge body of literature analyses symbolic sequences by means of Kolmogorov complexity, including diagnostic of the authorship of literary and musical texts (reviews in [17,22]). The measure is based on Kolmogorov complexity when evaluating the complexities of texts in different languages. It measures the information content of a string by the length of the shortest possible description required to (re)construct the exact string [76]. Although Kolmogorov complexity is uncomputable, it can be conveniently approximated with text compression programs. The basic idea behind the compression technique is that text samples which can be compressed comparatively better are linguistically comparatively less complex [77].

Lines 242-245: the author compares complete hunting stereotypes of a natural ant colony vs incomplete hunting stereotypes of laboratory-reared ants. What do “complete” and “incomplete” mean in this context? This distinction should be introduced in the text

Answer: I added the following fragment (lines 314-323) :

As the starting point of a hunting stereotype, the authors took the ant’s approach to the victim and the display of purposive movements; transportation of the killed victim was considered the end of the complete stereotype. All cases of loss of the victim and switching to another one were considered ends of incomplete stereotypes. It turned out that complete (successful) hunting stereotypes in members of a natural ant colony are characterised by smaller complexity than incomplete hunting stereotypes in naive laboratory-reared ants. Together with similar results obtained later on rodents [21,34], this application of Kolmogorov complexity could help to distinguish between innate and learned components of behavioural stereotypes.

Line 256: The author claims that this method can identify innate behaviours without mentioning any evidence to support this. Could you please elaborate on this point? What is the link between the complexity of a behaviour and its innateness?

Answer: I added the following fragment (lines 304 - 323:

Rearing ants from pupae in the laboratory enabled researchers to reveal that ants possess innate behavioural programs of different completeness. Those specimens equipped with the complete pattern enjoy an inherited template of potential prey and a readily available hunting stereotype; others possess fragments of the pattern and complete them through simple forms of social learning [17,34]. Ryabko et al. [17] compared two groups of highly genetically variable ants: members of a natural colony (“wild”) and naive (laboratory-reared from pupae) ants. They represented behavioural sequences as texts in which behavioural units (10 in total), singled out from video records and denoted by symbols (letters), served as an alphabet and tested the Hypothesis H0 (the sequences from two sets are generated by one source) against H1 (the complexity of sequences from one set is, on average, larger than the complexity of sequences from the other). As the starting point of a hunting stereotype, the authors took the ant’s approach to the victim and the display of purposive movements; transportation of the killed victim was considered the end of the complete stereotype. All cases of loss of the victim and switching to another one were considered ends of incomplete stereotypes. It turned out that complete (successful) hunting stereotypes in members of a natural ant colony are characterised by smaller complexity than incomplete hunting stereotypes in naive laboratory-reared ants. Together with similar results obtained later on rodents [21,34], this application of Kolmogorov complexity could help to distinguish between innate and learned components of behavioural stereotypes.

Line 276: As it is described, this method seems sensitive to the presence of regularities/stereotypes within 'texts' representing behavioural sequences. As the author pointed out, it is useful to measure their relative complexity. How can this method help to determine similarities in behavioral strategies of animals, too? Does it concern any analysis of the content (not just the regularity) of the behavioural sequence?

Answer: I consider the method that serves this purpose in a separate section. I added the following fragment (lines 371-374.) to clarify this:

As described in the previous section, the data-compression method is sensitive to the presence of regularities within texts representing behavioural sequences and effective to measure their relative complexity. However, this method does not help to determine similarities in behavioural strategies of animals.

Line 340: “However, hamsters and rats are not phylogenetically close, possibly caused by the particular abilities of these three species to manipulate with forepaws when handling the prey.” Could you please rephrase this line? The subject of the verb “caused” is not clear.

Answer: I changed this sentence: ...hamsters and rats are not phylogenetically close, possibly caused by the particular abilities of these three species to manipulate with forepaws when handling the prey.

Line 500: “Interestingly, the quantitative characteristics of the ants’ “number system” seem to be close, at least outwardly, to some archaic human languages: the length of the code of a given number is proportional to its value.” Could you please elaborate on this? It is not clear what you mean and which evidence can support this statement.

Answer: I changed this fragment as follows (lines 598 - 606):

For example, the word “finger” corresponds to 1, “finger, finger” to the number 2, “finger, finger, finger” to the number 3 and so on [134,135]. Of course, this is nothing ore than a superficial analogy. The numerical meaning attached to fingers is culturally encoded and in strikingly diverse ways [136]. In modern human languages, the length of the code word of a number j is approximately proportional to log j (for large j), and the modern numeration system results from a long and complicated development. However, the concept of a linear number line appears to be a cultural invention that fails to develop in the absence of formal education [135] (see review in [137]).

Line 513: “The main idea here explores the fundamental statement of information theory: in a “reasonable” communication system, the frequency of usage of a message and its length must correlate. The informal pattern is quite simple: the more frequently a message is used in a language, the shorter the word or the phrase coding it. This phenomenon is manifested in all known human languages.” Could you provide any reference to support this?

Answer: I changed this fragment as follows (lines 613 - 619): The main idea is to exploit the fundamental statement of information theory known as the Zipf’s law [140]: in a “reasonable” communication system, the frequency of usage of a message and its length must correlate. The informal pattern is quite simple: the more frequently a message is used in a language, the shorter the word or the phrase coding it [141]. This phenomenon is manifested in all known human languages [142] and recently has been analysed in animal communication [115].

Line 518: Could you please describe this experiment more accurately? I think the main confusion arises between the words “regularity” and “probability”, “stage” and “series”. It is not clear if different stages comprised different series and trials and if the same individual performed all the stages. If the same individual performed different trials: “the probability of finding the reward on a particular branch was 1/30”, does it mean that the probability for an individual to find the rewarded branch is 1/30, or that that specific branch was rewarded 1 out of 30 trials? What is a “series” of trials and which stage contained more than one series? If the same individual performed all the stages, then the difference between the three stages can be just a learning effect. I am not doubting the methods of this experiment (which has already passed the peer review). I am just giving to the author some feedback about the general confusion that a potential reader has while reading this part of the manuscript.

Answer: to meet this concern, I extended this section and added some explanations and clarifying details:

(1) lines 503-535:

To reveal the power of ants’ “language,” the researchers applied two central notions of information theory, that is, (1) the quantity of information and (2) the duration of time spent by the agents for transmitting 1 bit. The crucial idea is that the experimenters know exactly the quantity of information to be transferred. This approach, which is based on the “binary tree” experimental paradigm [19], enabled the authors to estimate the rate of information transmission in ants and to reveal that these intelligent insects can grasp regularities in the "texts" (such as LRRL... where L is “left” and R is “right”) and use them for coding and “compression” of information. These abilities may be considered the most complex properties of ants and animal cognition and communication in general [31]. The other series of experiments on “counting ants” was based on the Shannon’s equation connecting the length of a message (l) and its frequency (p), i.e., l = −log p, for rational communication systems. Applying this concept, Reznikova and Ryabko [23] demonstrated that ants could transfer information about the number of objects to each other and even add and subtract small numbers to optimise their messages. Below I explain some details of these experiments.

A general scheme of all three series of long-term experiments (“binary tree,” “counting,” and “arithmetic”) was based on the necessity to transfer distant information from scouting ants to members of their foraging groups to obtain food (see details in [7,8,37,90]). Each laboratory ant colony lived in an arena, in a transparent nest that made it possible for their activity to be observed. The arena was divided into two sections: the smaller one containing the nest, and the bigger one with an experimental system (Fig. 2).bath of water. All actively foraging ants were individually marked with paint. In each series or experiments, several hundred of scouting ants and their foraging groups took part. For example, 335 scouts and their teams from three F. polyctena colonies were used in the experiments with the binary tree. The scouts and their teams took part in different

trials switching from one task to another. In each trial, the experimenter placed one of the scouts on a definite point of a maze, for example, on a leaf of the binary tree (or a branch of a counting maze) that contained a trough with the food, and then it returned to the nest by itself. There were no cues that could help the ants find the food (including olfactory ones) except the information contacts with scouts (see the detailed description, tables and figures in [72,107,122]). When a scout returned home, the experimenter measured the duration of its contact with foragers in seconds.

(2) lines 619-623:

The main experimental procedure in this “arithmetic” series of experiments was similar to other experiments with “counting mazes” [19,23] Four colonies of red wood ants took part in experiments over three years, with the number of trials ranging from 92 to 150 in different years.

(4) Lines 672-681:

It can be suggested that the scheme of experiments based on solving cooperative tasks can be directly used on some non-human mammals and young human infants. For example, pairs or small groups of subjects can be involved. In this case, one subject is shown the container with reward but has to communicate its location to the other for both of them to get it. The reward can be placed in one out of (say) 20 containers, which play the same role as the branches of the comb-like maze in the experiments with ants. The numerical abilities of the subjects can then be studied by analysing the time the first subject (that has seen the reward) spends on transferring the information on its location to the other, precisely as in the experiments with ants.
